# Anomalous and Chern topological waves in hyperbolic networks

Qiaolu Chen[1,2,3], Zhe Zhang [1,3], Haoye Qin[1], Aleksi Bossart[1], Yihao Yang [2], Hongsheng Chen [2] & Romain Fleury [1] ✉

Hyperbolic lattices are a new type of synthetic materials based on regular tessellations in non-Euclidean spaces with constant negative curvature. While so far, there has been several theoretical investigations of hyperbolic topological media, experimental work has been limited to time-reversal invariant systems made of coupled discrete resonances, leaving the more interesting case of robust, unidirectional edge wave transport completely unobserved. Here, we report a non-reciprocal hyperbolic network that exhibits both Chern and anomalous chiral edge modes, and implement it on a planar microwave platform. We experimentally evidence the unidirectional character of the topological edge modes by direct field mapping. We demonstrate the topological origin of these hyperbolic chiral edge modes by an explicit topological invariant measurement, performed from external probes. Our work extends the reach of topological wave physics by allowing for backscattering-immune transport in materials with synthetic non-Euclidean behavior.

Non-Euclidean geometry assumes non-zero spatial curvature, thereby deviating from the axioms and postulates of standard Euclidean geometry. Its advent in the early 19th century marked a major turning point in comprehending the universe. An important example is the hyperbolic geometry[1], which is relevant in general relativity to represent gravitational effects as the geometric curvature of space–time, and furnishes an avenue to understand the "event horizon" of a black hole. Beyond this, hyperbolic geometry also has substantial applications in mathematics, computer graphics and information theory. As a result, there has been a growing interest in visualizing hyperbolic geometry in Euclidean pictures (see Fig. 1a) and exploring theoretically and experimentally the non-trivial physics of hyperbolic spaces.

Discrete translational symmetry supported in Euclidean lattices is a cornerstone of modern physics, as it enables the propagation of Bloch waves associated with a commutative translation group in Euclidean spaces. In analogy with the Euclidean case, hyperbolic lattices that exhibit discrete periodicity in hyperbolic spaces can also support Bloch waves but with a twist: they belong to a non-commutative group of hyperbolic translations, the Fuchsian group

$\Gamma \subset PSU(1,1)$[2–8]. This complexity therefore results in distinct spectra[9] and band topology in hyperbolic momentum space, as theoretically proposed in the hyperbolic analogues of the quantum spin Hall effect[10], Chern insulator[11], higher-order topology[12], Haldane[13] and Kane–Mele models[5]. Experimental demonstrations, all of which, however, are implemented in reciprocal systems, have been conducted in circuit quantum electrodynamics[14] and topolectrical circuits[15–18]. These lumped circuits preserve time-reversal symmetry, making it impossible to probe robust topological phases associated with non-reciprocal edge modes, such as the Chern[19–26] and anomalous Floquet insulators (both belonging to class A)[27–32], the latter being the most robust two-dimensional topological phase known[31,32]. We notice that a recent work[17] simulates an infinite hyperbolic lattice, where the periodic boundary condition with Bloch phases breaks the time-reversal symmetry and reciprocity, as expected for any Bloch boundary condition. However, it is applied to an infinite hyperbolic lattice that does not break time-reversal symmetry or reciprocity. In addition, these lumped circuits operate in a quasi-static regime governed by Kirchhoff's laws, describing a discrete system with no spatial extent in

[1]Laboratory of Wave Engineering, School of Electrical Engineering, EPFL, Lausanne, Switzerland. [2]Interdisciplinary Center for Quantum Information, State Key Laboratory of Modern Optical Instrumentation, ZJU-Hangzhou Global Science and Technology Innovation Center, College of Information Science and Electronic Engineering, ZJU-UIUC Institute, Zhejiang University, Hangzhou, China. [3]These authors contributed equally: Qiaolu Chen, Zhe Zhang. ✉e-mail: romain.fleury@epfl.ch

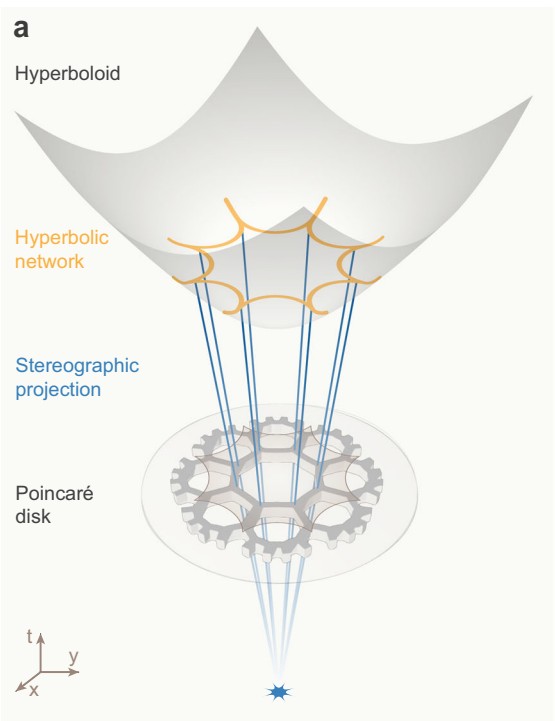

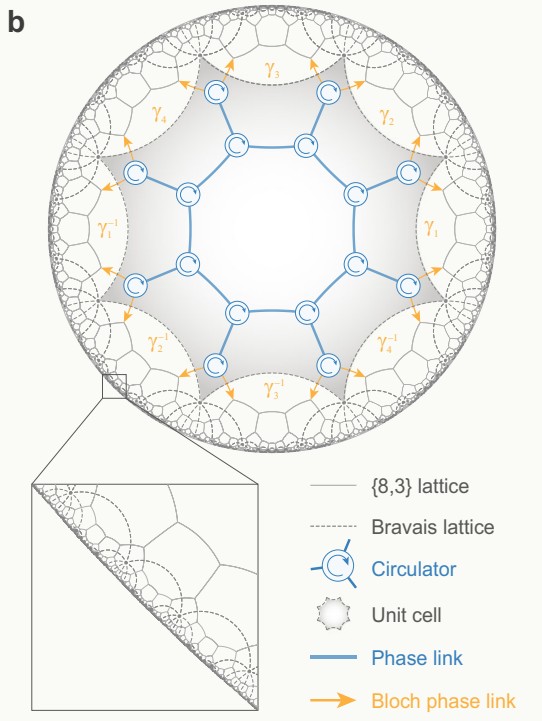

**Fig. 1 | Hyperbolic non-reciprocal scattering wave network. a** A periodic lattice (orange lines) on a hyperboloid ($t^2-x^2-y^2=1$) in (2+1)-dimensional ($x, y, t$) Minkowski space is mapped onto a Poincaré disk ($t = 0$) by stereographic projection through the point (0, 0, −1). These two models of the hyperbolic lattice are isometrically equivalent. **b** Poincaré disk model of a {8,3} hyperbolic lattice in a non-reciprocal scattering wave network. Here, the grey solid lines represent the {8,3} hyperbolic lattice, and the grey dashed lines are the corresponding {8,8} hyperbolic Bravais lattice. A central unit cell is highlighted in shaded grey, which consists of 16 three-port non-reciprocal circulators connected via bidirectional phase links (blue solid lines) (see a detailed unit cell in Supplementary Fig. 1b). Adjacent unit cells are connected by the Bloch phase links (orange arrows).

which, by definition, wave behavior is absent. Thus, the observation of robust wave transport phenomena in topological hyperbolic lattices has remained elusive thus far.

In this work, we design and demonstrate experimentally a topological hyperbolic lattice in a non-reciprocal scattering wave network, establishing unidirectional channels to induce new and exciting transport properties in curved spaces. Taking advantage of the non-reciprocity of the scattering wave network, we unveil the existence of two distinct classes of chiral topological edge modes, the Chern and anomalous types, whose hyperbolic topology is confirmed by a Chern vector and a unitary homotopy invariant. We observe them experimentally for electromagnetic waves in the GHz range and validate the unidirectional character of their propagation. An experimental proof of the topological nature of the anomalous topological edge modes is provided by directly measuring a topological invariant based on a generalization of Laughlin's pump argument[33] to hyperbolic samples.

## Results

### A non-reciprocal hyperbolic scattering wave network

Scattering wave networks are systems in which waves travel along the phase delay links of a complex graph, whose nodes correspond to scattering events. They have been exploited as simple yet powerful platforms for investigating topological wave physics in Euclidean spaces. One important achievement in these networks is the realization of the anomalous edge mode, a chiral topological mode that can be found when node scattering exhibits sufficiently low reflection and non-reciprocity[34,35]. Compared to the traditional case of Chern edge modes, anomalous edge modes are robust to any distributed disorder in the structure of the planar graph, with superior resistance to backscattering. However, no graph is allowed on an Euclidean plane: a simple regular octagon tiling, for example, requires negative curvature. The extension of such chiral edge modes to hyperbolic spaces is thus by no means trivial.

Exploring this possibility, we implement a hyperbolic lattice in a planar non-reciprocal scattering network by decoupling the physical distances from the metric and distorting a curved hyperbolic tessellation onto a two-dimensional Poincaré disk, as shown in Fig. 1b. The hyperbolic lattice is constructed by regular octagons, where three octagons meet at one node (lattice site), building a {8,3} hyperbolic lattice. In our scattering wave network, the scattering nodes (or lattice sites) are three-port non-reciprocal unitary scatterers with $C_3$ symmetry (i.e. circulators), linked with bidirectional connections involving a phase delay $\varphi$. The scattering process at each node can be generically described by a $3 \times 3$ asymmetric unitary scattering matrix $S_0$, whose general parametrization involves only two angles, $\xi$ and $\eta$, each ranging from $-\pi/2$ to $\pi/2$ (see Supplementary Note 1). By varying these two angles, we can explore a family of circulators with all possible degrees of non-reciprocity and reflection.

A recently developed hyperbolic Bloch theory reveals that, despite the non-commutative character of its translation groups, the hyperbolic lattice admits a momentum-space description based on the notion of a hyperbolic Bravais lattice and a higher-dimensional Brillouin zone[7]. Following this insight, our {8,3} hyperbolic lattice is associated to a {8,8} hyperbolic Bravais lattice (see Fig. 1b). The corresponding unit cell is an octagon containing 16 lattice sites, equipped with a Bloch vector $\mathbf{k} = (k_1, k_2, k_3, k_4)$ under a non-commutative lattice translation group $\Gamma = \langle \gamma_1, \gamma_2, \gamma_3, \gamma_4 : \gamma_1 \gamma_2^{-1} \gamma_3 \gamma_4^{-1} \gamma_1^{-1} \gamma_2 \gamma_3^{-1} \gamma_4 \rangle$. These momentum components form a four-dimensional Brillouin zone, whose dimensionality is determined by the minimal number of hyperbolic translation generators. As a consequence, one can establish a Bloch eigen-equation for the hyperbolic scattering wave network

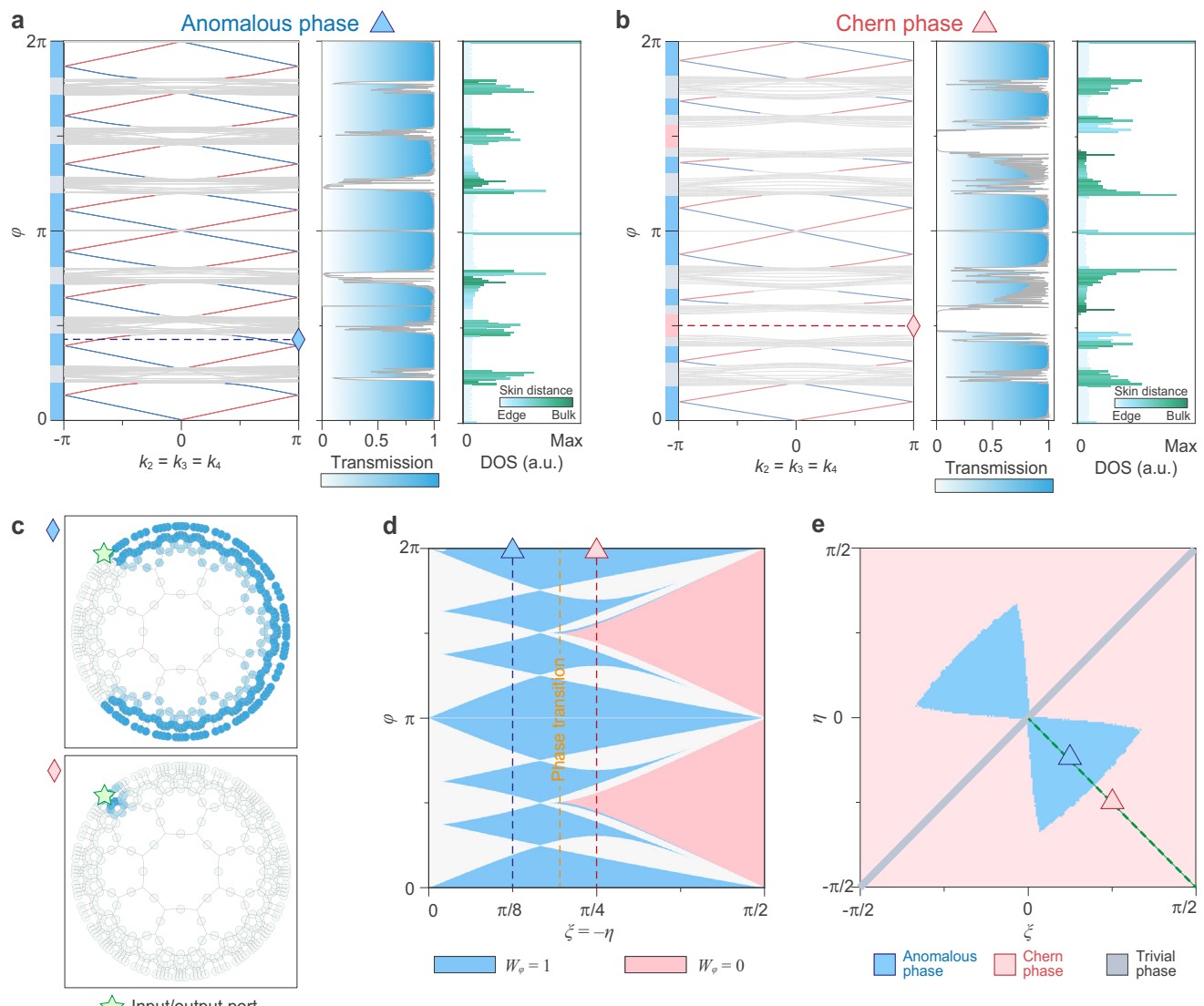

**Fig. 2 | Chiral edge modes in hyperbolic anomalous and Chern phases. a** and **b** Floquet band structures (leftmost panels), two-port edge transmissions (middle panels), the density of states and skin distance (rightmost panels) for an anomalous phase (panel **a**, $\xi = -\eta = \pi/8$), and a Chern phase (panel **b**, $\xi = -\eta = \pi/4$). It shows that the hyperbolic Bloch band theory and the exact diagonalization calculation give a complete band characterization of hyperbolic lattices, confirming the anomalous phase in panel a with chiral edge modes filling every quasi-energy bandgap, and the Chern phase in panel **b** with two additional trivial bandgaps. **c** Simulated wave propagation in the hyperbolic scattering network, showing an anomalous chiral edge mode in a topological bandgap ($\varphi = 2\pi/5$) and no mode with total reflection in a trivial bandgap ($\varphi = \pi/2$). **d** Topological bandgap map on $\xi = -\eta$ line determined by homotopy invariant $W_\varphi$, with $W_\varphi = 1$ for topological bandgaps (blue areas) and $W_\varphi = 0$ for trivial bandgaps (red areas). The grey areas represent the bulk bands. A phase transition occurs at $\xi = -\eta = 11\pi/60$. **e** Topological phase diagram on ($\xi, \eta$) plane. Blue and red areas correspond to anomalous and Chern phases. Grey line ($\xi = \eta$) represents a trivial insulating phase.

(see Supplementary Note 2)[31],

$$S(\mathbf{k})|b(\mathbf{k})\rangle = e^{-i\varphi(\mathbf{k})}|b(\mathbf{k})\rangle. \qquad (1)$$

Here, $S(\mathbf{k})$ is a $48 \times 48$ unitary scattering matrix describing the scattering property of circulators within a unit cell and the connectivity between them. Due to the unitarity of $S(\mathbf{k})$, the eigenvalue spectrum $e^{-i\varphi(\mathbf{k})}$ lies on a unit circle, defining a Floquet eigen-problem with quasi-energy $\varphi(\mathbf{k})$ confined within this compact space, i.e. $\varphi(\mathbf{k}) \in [0, 2\pi)$.

**Chiral edge modes in hyperbolic anomalous and Chern phases**

The model of Eq. (1) provides a description of the hyperbolic scattering wave network in terms of a Floquet band structure and Bloch-wave functions, which we use to explore the hyperbolic topological physics. We numerically compute the Floquet band structures for a supercell, with periodic boundary conditions along the $k_2$, $k_3$, and $k_4$ directions and

full-reflection boundary conditions along $k_1$. Figure 2a (leftmost panel) depicts the obtained band structure for circulator parameters $\xi = -\eta = \pi/8$, corresponding to a low-reflection case. The band structure exhibits eight bandgaps filled with chiral edge modes, which are located on two opposite sides of the boundary (see Supplementary Fig. 5b). While the computed Chern vectors vanish for all bands (see Supplementary Note 4 and Supplementary Fig. 8a), the unitary homotopy invariant $W_\varphi$[27,31,34] confirms the non-trivial topology of all gaps. We find that they are all topological with $W_\varphi = 1$ (blue areas on the left side of band structure), defining a hyperbolic extension of the anomalous Floquet phase[27,29,31,34,36]. The situation is different at a higher value of the circulator reflection ($\xi = -\eta = \pi/4$). As shown in Fig. 2b (leftmost panel), apart from eight topological bandgaps, the band structure exhibits two additional trivial bandgaps with homotopy invariant $W_\varphi = 0$, surrounded by bands with non-zero Chern vectors (see Supplementary Fig. 8b), namely the hyperbolic version of a Chern insulator.

To test for hyperbolic topological transport associated with the chiral edge modes, we compare the Floquet band structure in the thermodynamic limit to the edge transmission computed for a finite network. The finite network is made of $n = 3$ layers of neighbouring octagons with two probes placed on the boundary (all other boundary ports being closed with full-reflection boundary conditions). Figure 2a and b (middle panels) represent the edge transmission versus quasi-energy $\varphi$. As expected, the transmission in a topological (trivial) bandgap approaches one (zero) and fluctuates within bulk bands. We further validate this observation by visualizing the wave propagation in the network. As shown in Fig. 2c, wave transport is unidirectional along the boundary (top panel: case of an anomalous topological bandgap with $\varphi = 2\pi/5$). For comparison, we include the case of a trivial bandgap of the Chern phase (bottom panel: $\varphi = \pi/2$), for which waves are totally reflected.

In addition, we provide another point of view based on the study of the density of states (DOS)[37] and skin distance in a closed, finite hyperbolic network. The DOS is defined as $DOS(\varphi) = N(\varphi)/\delta(\varphi)$, where $N(\varphi)$ counts the number of eigenmodes of the closed network within $[\varphi, \varphi + \delta(\varphi)]$. The skin distance is defined as $p_s(|b\rangle) = \sum_i p_s(i)|b_i|^2/\sum_i |b_i|^2$, where $|b_i|$ is the amplitude of mode $|b\rangle$ at port $i$, and $p_s(i) \in \mathbb{N}$ is the skin index of port $i$, ranging from $p_s = 1$ on the boundary to a (size-dependent) maximal value at the center. The skin distance thus captures the averaged position of an eigenmode. The results in the rightmost panels of Fig. 2a and b show that when $\varphi$ falls in a topological bandgap, the spectrum is dominated by edge modes, with DOS and skin distance reaching low values. Conversely, if $\varphi$ belongs to a trivial bandgap, DOS is zero, and skin distance is undefined. Finally, if $\varphi$ falls in a bulk band, both DOS and skin distance reach high values depending on $\varphi$, indicating diffusive waves in bulk. With the above topological invariants and observables, we, therefore, conclude that hyperbolic scattering wave networks can indeed support both anomalous and Chern phases, with chiral topological edge modes in every bandgap (anomalous phase), or only in some of them (Chern phase).

Figure 2d displays the topological bandgap map of anomalous and Chern phases along the $\xi = -\eta$ angular parameter line. For a given value of $\xi = -\eta$, the hyperbolic network is either in a topological bandgap ($W_\varphi = 1$, blue area), a trivial bandgap ($W_\varphi = 0$, red area), or a bulk band (grey area). Besides, a phase transition occurs at $\xi = -\eta = 11\pi/60$, with anomalous and Chern phases on its left and right sides, respectively. We also map out in Fig. 2e the complete topological phase diagram for every possible value of $(\xi, \eta)$. The red and blue areas correspond to Chern insulators and anomalous Floquet insulators. The grey line represents the trivial insulating phase, where its middle point ($\xi = \eta = 0$) corresponds to a semi-metallic phase with all bandgaps closed, serving as a crossing point of three phases (i.e., anomalous, Chern, and trivial phases).

## Observation of hyperbolic chiral transport

We design and build a prototype of a hyperbolic non-reciprocal scattering network, which is composed of ferrite circulators linked with coplanar waveguides (CPWs) and operating at microwave frequencies (see Supplementary Note 7). To distort the curved tiling of regular octagons onto the flat Euclidean plane, we adjust flexibly the physical layout of each CPW while keeping its length unchanged, such that the periodic nature of the lattice is guaranteed. Such flexibility allows the arrangement of a curved lattice on a flat platform. A picture of the prototype is shown in Fig. 3a. We note that, to accommodate the CPWs on the extensive boundary and guarantee all the CPWs experience approximately the same phase delay within the desired frequency range of interest, the CPWs forming the central octagon are adjusted to be longer than the others. These longer CPWs exhibit an additional $2\pi$ phase delay at 5.93 GHz, effectively ensuring the phase difference remains within $\leq 5\%$ across the entire frequency

range of [5.75, 6.10] GHz (see Supplementary Fig. 13d). Aside from the lattice itself, the network holds eight input/output ports for measurements.

The first step is to extract the scattering matrix for a single circulator, which predicts the hyperbolic scattering network stands in an anomalous phase in the frequency range [5.5, 6.5] GHz and in a Chern phase in [5.0, 5.5] GHz (see Supplementary Fig. 12a). Taking into account the dispersion of CPWs, we could also predict the bandgap map in the frequency spectrum, as shown in Supplementary Fig. 12d. Next, we measure the field distributions by exciting an input port on the boundary and probing the fields manually at each CPW. Figure 3e reports the wave propagation between 5.8 and 6.1 GHz, visualizing the anomalous chiral edge modes transport clockwise along the boundary while the bulk modes scatter in all possible directions. We note that these bulk modes exhibit non-reciprocity to some extent but in a diffusive manner, as they typically consist of a mixture of modes given the high DOS.

## Measurement of a topological invariant

To probe experimentally the topology of our hyperbolic lattice, we perform a topological pumping experiment allowing for the direct measurement of a scattering topological invariant by generalizing Laughlin's argument to the hyperbolic case[33,38–40]. The procedure consists of first cutting the planar network model with Corbino disk geometry along the radial direction[41], and second, rolling it into a truncated cone by bridging the gap using twisted boundary conditions (see Fig. 3b). The twisted boundary conditions impart a non-reciprocal phase delay, which serves as a synthetic magnetic flux $\Phi$ threading the truncated cone. One then defines the topological invariant $W$ in hyperbolic scattering networks as the winding number of the probe reflection coefficient $R$ on the bottom edge during an adiabatic cycle of the flux $\Phi$[32,42–44], namely, $W = 1/(2\pi i) \int_0^{2\pi} d\Phi\, R^*(\partial R/\partial \Phi)$ (see Supplementary Note 5 and Supplementary Fig. 10).

In practice, the synthetic magnetic flux is implemented with reconfigurable non-reciprocal phase shifters that provide a non-reciprocal phase delay $\pm \Phi$, controlled by external voltages $V_1$ and $V_2$ (see Supplementary Fig. 14). The twisted boundary condition is then applied by inserting three phase shifters within the cut links, as shown in Fig. 3c. The measured topological invariant in the frequency spectrum is reported in Fig. 3d, showing $W = 1$ in frequency ranges [5.8, 5.82] and [5.98, 6.07] GHz, and $W = 0$ in other ranges. This is in perfect agreement with the predicted bandgap map obtained from the measured scattering matrix $S_0$ of the single circulator. To complete the picture, we map out several measured windings of reflection coefficient on the complex plane, as shown in Fig. 3f. As expected, the reflection coefficient at 5.8 and 6.0 GHz winds in a non-contractible loop. Such a non-zero winding constitutes strong evidence of the topological non-triviality of the anomalous chiral edge modes. The situation is different for bulk modes at 5.9 and 6.1 GHz, whose reflection coefficient does not exhibit any winding. The matching between theoretical predictions, measured field maps, and measured topological invariants is excellent, confirming the existence of anomalous edge transport in our prototype.

## Discussion

We have experimentally proven that non-reciprocal scattering networks represent an ideal platform to explore topological wave physics in a curved hyperbolic lattice, giving access to substantial physical implications and potential applications of non-Euclidean geometry in wave transport and control. Taking advantage of non-reciprocity, hyperbolic chiral transport has been demonstrated by expanding the celebrated notions of Chern and anomalous phases to the hyperbolic world. These distinct phases can, for example, be used to reconfigure a domain wall between two differently tessellated hyperbolic networks and redirect the hyperbolic energy flow without flipping the magnetic

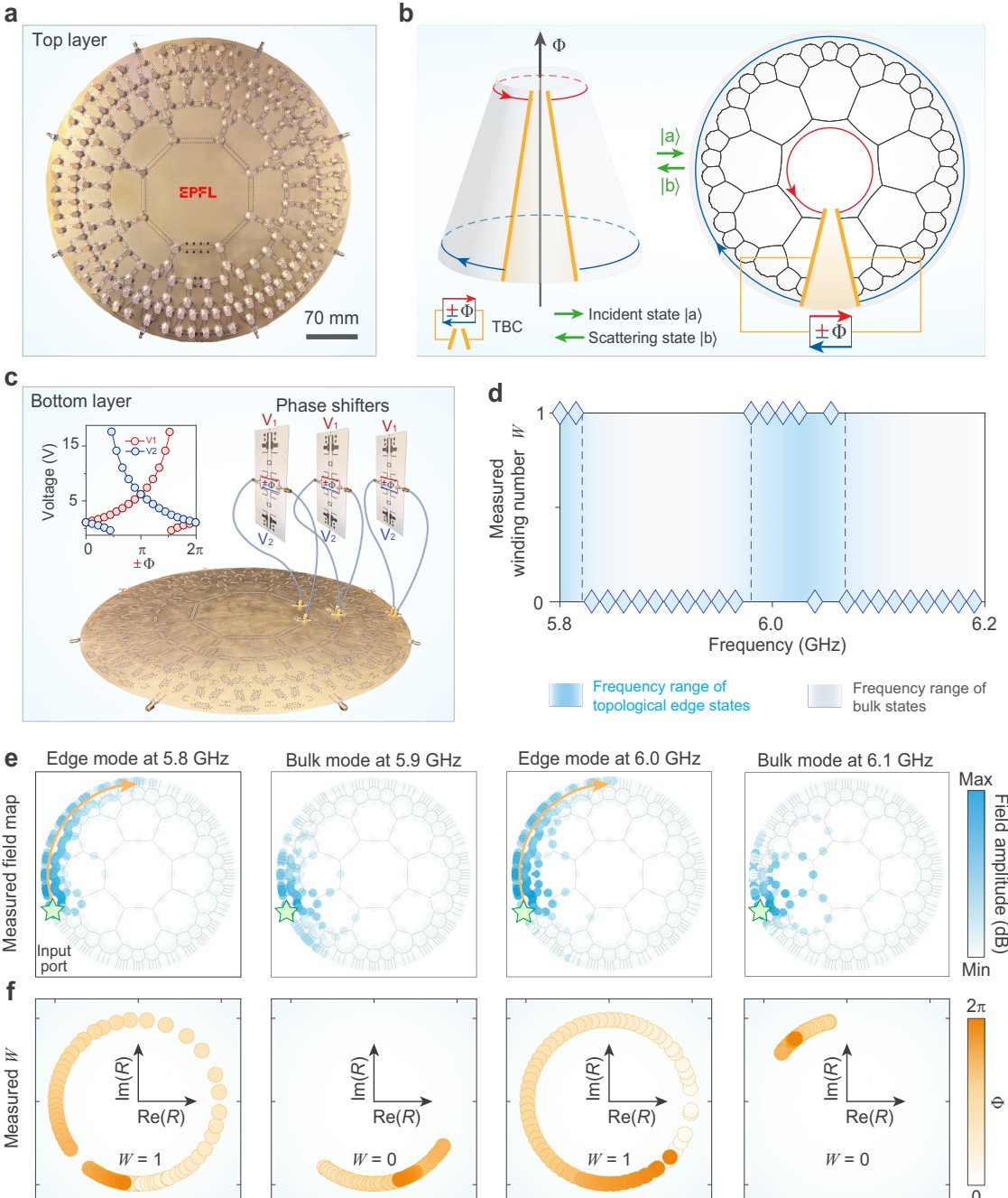

**Fig. 3 | Experimental demonstration of topological transport in hyperbolic networks. a** Photograph of the experimental prototype (top layer), consisting of 200 circulators with $n = 3$ layers. The circulators are connected by coplanar wave-guides (CPWs). **b** Schematic for measuring the topological invariant of the hyperbolic lattice. The magnetic flux $\Phi$ in the topological pump (left panel) is synthetically realized by applying twisted boundary conditions with a tunable non-reciprocal phase delay $\pm\Phi$ (right panel). The topological invariant $W$ counts the winding of the reflection coefficient at a probe on the edge during a cycle of the non-reciprocal phase delay from 0 to $2\pi$ (from 0 to $-2\pi$ in the opposite direction). **c** Experimental setup for measuring the topological invariant in the hyperbolic scattering network. Three designed phase shifters provide phase delays $+\Phi$ and $-\Phi$ in two opposite directions by tuning the bias voltages $V_1$ and $V_2$, respectively. The inset depicts the phase delay as a function of bias voltage. **d** Measured topological invariant $W$ in the frequency spectrum. The darker blue and grey areas (split by dashed lines) respectively feature edge modes and bulk modes, according to the predicted bandgap map (see Supplementary Fig. 12d). **e, f** Measured field maps (**e**) and measured windings of reflection coefficient (**f**). The frequencies are 5.8 GHz ($W = 1$, edge mode), 5.9 GHz ($W = 0$, bulk mode), 6.0 GHz ($W = 1$, edge mode), and 6.1 GHz ($W = 0$, bulk mode).

field. Anomalous hyperbolic transport, in particular, may form a basis for robust point-to-point energy transport in non-Euclidean spaces. Generalizing Laughlin's pump argument to hyperbolic geometries[33], we have provided conclusive evidence for the topological origin of the hyperbolic chiral transport. Such a practical method is general and applicable for arbitrary scattering networks in both Euclidean and non-Euclidean lattices. Compared to Euclidean networks, hyperbolic networks exhibit a larger boundary-bulk ratio, allowing for the creation of chiral transport over a long edge, albeit with a compact bulk. We surmise that exploring the implications of bulk-boundary correspondence in other types of hyperbolic spaces, or in the presence of non-Hermiticity, will soon unveil other surprising edge or skin

phenomena[45,46] based on the interplay between topology, non-Hermiticity, and negative spatial curvature.

## Methods

More extensive information, along with additional data and discussion regarding theory, simulations, and experiments, can be found in the supplementary information file.

## Data availability

The data that support the findings of this study are available at https://zenodo.org/records/10409107.

## Code availability

The codes that support the findings of this study are available at https://zenodo.org/records/10409107.

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

## Acknowledgements

This work was supported by the Swiss State Secretariat for Education, Research and Innovation (SERI) under contract number MB22.00028 (R.F.) and the Swiss National Science Foundation under the Eccellenza award 181232 (R.F.). We thank Pierre Delplace, Yifei Guan, and Zhechen Zhang for their helpful discussions.

## Author contributions

R.F. and Z.Z. conceived the project. Q.C. and Z.Z. carried out the analytical and numerical modelling. Q.C. designed and fabricated the sample. Q.C. conducted the measurements with the assistance of H.Q. Q.C. performed data analysis. Q.C. wrote the manuscript. Q.C., R.F., Z.Z., A.B., Y.Y. and H.C. revised the manuscript. R.F. supervised the entire project. All authors discussed the results and commented on the article.

## Competing interests

The authors declare no competing interests.
