## [Peer Review File · Nature Communications]

REVIEWER COMMENTS

Reviewer #1 (Remarks to the Author):

The authors present a theoretical + experimental study of a non-reciprocal microwave scattering network with the geometry of the {8,3} hyperbolic lattice. Each node of the lattice supports a non-reciprocal “circulator” element which breaks time-reversal symmetry and allows for the possibility of topological phenomena such as Chern numbers and chiral edge states.

On the theoretical side, the authors first model an infinite scattering network using hyperbolic band theory. The system is defined by a unitary but not symmetric (on account of time-reversal breaking) S-matrix $S(k)$ where k lives in a higher-dimensional Brillouin zone. The eigenvalues of this S-matrix play a role analogous to the quasi-energy band structure of a periodically driven (Floquet) system. Depending on the range of parameters for the circulators, this band structure supports band gaps whose topology can be characterized by momentum-space topological invariants, either Chern numbers or winding numbers discussed in the Floquet context. From the momentum-space analysis, the authors predict “anomalous Floquet insulator” phases with nontrivial winding numbers (but trivial Chern numbers) and Chern phases with nontrivial Chern numbers. The authors then investigate edge phenomena by imposing open (reflecting) boundary conditions in one direction in the band theory, or directly simulating a finite lattice with an edge. They find that band gaps with nontrivial winding number (not momentum-space Chern number) are associated with chirally dispersing modes in both the band theory and the real-space simulation.

Next, the authors test these predictions experimentally by both measuring edge propagation and also directly measuring the “bulk” topological invariant (winding number) directly. The latter is measured by introducing specific tunable phase elements that impose a phase difference across a “cut” through the device, and measuring the corresponding winding in the reflection coefficient, which detects spectral flow across the gap in the scattering resonances over a flux-pumping period.

In my opinion, this is a careful and detailed study that constitutes a significant advance in the field of hyperbolic lattices. From the experimental standpoint, it introduces microwave scattering networks as a new physical platform to study non-Euclidean phenomena in condensed matter physics, alongside circuit QED architectures and (low-frequency) electric circuit networks. It also investigates non-reciprocity in the hyperbolic context in the first time (although Ref. 17 did introduce time-reversal breaking complex phase elements), and demonstrates in a compelling manner the presence of chiral topology through both bulk and edge measurements. From the theoretical standpoint, it extends the study of topological phenomena in hyperbolic lattices to scattering networks.

I am happy to recommend the paper for publication but would first like the authors to address the following questions/comments.

1. The authors state that previous experimental work has left “unidirectional edge wave transport completely unobserved” in the hyperbolic context. However, Ref. 16 claims to have experimentally realized a hyperbolic version of the Haldane model and observed unidirectional propagation of a voltage package on the edge. The authors should better clarify the relationship of this work to theirs.

2. Likewise, it would be useful to have a brief discussion of time-reversal breaking vs non-reciprocity, given the experiment of Ref. 17 which had previously introduced complex phase elements in the hyperbolic context.

3. In the last sentence of the section “Chiral edge modes in hyperbolic anomalous and Chern phases”, the authors write that “the grey line on the center ($\xi=\eta$) denotes a trivial insulator with all bandgaps closed.” If the bandgaps are closed how can this be an insulator? Do the authors mean a metallic or semi-metallic phase instead?

4. At the end of the next section (Observation of hyperbolic chiral transport), the authors say that “the bulk modes scatter in all possible directions”. Yet from Fig. 3e it seems the scattering pattern is not completely isotropic but retains a bit of chirality. Do the authors have an explanation for this?

5. I eventually understood that the authors chose a longer CPW in the center of the device for purely geometric/engineering considerations (to have enough room to pack everything?) but I was initially confused by the two lengths. Although perhaps obvious to some, it might be useful to mention this point explicitly somewhere.

Reviewer #2 (Remarks to the Author):

The manuscript is an experimental study of the topological properties of models in non-Euclidean spaces. The Authors consider a planar projection of an hyperbolic lattice by coupling different scattering elements operating at microwave frequencies. In particular, the network connections between lattice sites in the hyperbolic space are realized via non-reciprocal circulators, where a phase delay is introduced. By varying two angular parameters of the circulator, setting different reflection and non-reciprocity degrees, a topological phase diagram is reconstructed. The findings are supported by the

observation of unidirectional propagation on the edge of the system, associated with nonzero topological invariants, measured through a generalized Laughlin pumping.

The manuscript advances the field of topological physics, reaching out to new synthetic metamaterials on hyperbolic lattices, where experimental investigation are just begun. I believe that these results can be of immediate interest to the community working in the field, as its importance can stretch towards novel interplay of topology and possibly also non-Hermitian effects in non-Euclidean spaces.

Overall the paper is well written and organized, while the methodology and the extended information are sufficiently detailed and provide enough support to the arguments presented in main text. However, I have few comments that the Authors should address in order to increase the clarity of the paper and its conclusions.

1) The band structure in Fig.2 is obtained with the phase of the circulator ϕ as a quasi-energy of a Floquet model. This analogy is formally evident in Eq.1 if the scattering matrix is the time-evolution operator of a temporally modulated Hamiltonian $H(t) = H(t+T)$ whose state is $|\psi(t+T)\rangle = e^{-i\epsilon T} |\psi(t)\rangle$, with ϵ being the quasi-energy. Is this analogy based on physical ground or is it just a formal one? What would be the temporal modulation of what quantity in your case?

2) Related to my previous comment, it is also not immediately evident to me why is the phase of the circulator ϕ used as energy in the band structure and not the actual energy/frequency. How would the plot Fig. 2d look like if the frequency was plotted instead of ϕ ? I expect some differences but at least the location of the gap-closing point signalling the topological phase transitions should exactly match.

3) Is the circulator breaking the time-reversal symmetry? I am thinking whether the Chern insulator phase belongs to class A or to some of the other 4-dimensional classes? Would be nice to have it written.

4) At page 5, when discussing the location of the chiral edge modes, it is said that they are located “either at the top edge (red lines) or the bottom one (blue lines)”. How are “top” and “bottom” edge defined here, since from Fig. 1 and 2c (but also Fig. 3e and so on) the real lattice is defined on a circle? I assume it has to do with the reflection boundary conditions along k_1 , but what is the meaning of propagation along k_1 in real space? It is hard to visualize otherwise.

5) At page 3, the Authors say “Compared to the traditional case of Chern edge modes, anomalous edge modes are robust to any distributed disorder in the structure of the planar graph, with superior resistance to backscattering.” I don’t think this sentence is true, as chiral modes are either protected or not protected, so a topological invariant cannot protect them “more” than another. Authors should

either rephrase the sentence, or provide evidence: for example, can this superior resistance to backscattering be seen from their experimental findings in Fig. 2 and 3?

6) The Authors use the Laughlin pumping argument to measure the winding number, and show that some bandgaps have zero winding for the Chern case, while the Chern number calculation is done via Wilson loop extraction. Would it be possible to measure some quantized bulk response to directly measure the Chern number?

Reviewer #3 (Remarks to the Author):

My comments are in the attached PDF.

Reviewer report on “Anomalous and Chern topological waves in hyperbolic networks”

In this manuscript, the authors designed and implemented a hyperbolic lattice in a planar non-reciprocal scattering network. They use hyperbolic band theory to compute band structure numerically and extracted the unitary homotopy invariant W_φ and Chern numbers from it. The numerical results show that their model could be in the anomalous phase or Chern phase depending on the parameter ξ and η and can support chiral edge waves in the anomalous phase. They further experimentally demonstrates the chiral edge waves and measured the unitary homotopy invariant W_φ based on the generalized Laughlin’s argument they developed for the hyperbolic lattice. This is a novel experimental systems on hyperbolic lattice with breaking time reversal symmetry and the authors showed for the first time chiral edge states in hyperbolic lattice. Given the novelty above, I recommend the paper be published in Nature Communication after the following comments has been addressed.

First, the authors computed the band structure of the model in 1D representations. However, there are two potential problems the authors need to explain due to the fact that the Fuchsian group has higher-dimensional irreducible representations and the hyperbolic lattice has a large boundary. First, the 1D irreducible representations only captures a small amount of bulk states, why can they give correct topological invariants that characterizes the boundary modes? Second, when cutting a finite piece from the hyperbolic lattice, the boundary to bulk ratio never tends to zero and the

boundary effect can not be ignored. It is important to check that the band structure can actually reflect the properties of the system. In other words, it is worthwhile to compute the spectrum of the real space scattering matrix and compare them to the band structure to see at least if the DOS is similar between the 1D band theory and the real space finite lattice.

Second, the authors mentioned that the chiral edge waves are robust in Euclidean lattices. Are they robust in hyperbolic lattice as well? If they are, can the authors add some numerical or experimental results demonstrate this?

Third, the authors numerically showed that their model can support Chern phase. However, they didn't show experimental phenomena associated to the non-trivial Chern vector. Can the author comment on why this part of the experiment is absent?

Fourth, in page three paragraph 3, the authors mentioned that "These momentum components form a four-dimensional Brillouin zone, whose dimensionality is determined by the number of hyperbolic translation generators." This statement is inaccurate as the number of hyperbolic translation generators are not well defined for general hyperbolic lattice: the Fuchsian group, being a non-abelian group, can have different generating set with different number of generators. The precise statement should be "These momentum components form a four-dimensional Brillouin zone, whose dimensionality is determined by the **minimal** number of hyperbolic translation generators."

Fifth, the φ in Eq.(1) depends on \mathbf{k} . It's better to explicitly mark this to avoid confusion.

Response letter, manuscript number NCOMMS-23-46090-T

We thank all three Referees for their careful review and positive comments of our manuscript. We appreciate, in particular, the Reviewers' keen interest in our work as evident from the detailed comments about the underlying physics and the strong potential of our manuscript to be a widely-read, timely contribution to the multidisciplinary field of non-Euclidean and topological materials.

The Reviewers have raised a number of questions that have helped us to (i) significantly improve the presentation of our findings and stress their significance; and (ii) perform new studies to better support the key conclusions that we draw. The two principal changes that we have made are:

- We now provide **additional statistical evidence** of the superior robustness of anomalous hyperbolic network to fully random phase disorder, compared with Chern hyperbolic network.

- We now demonstrate that most bulk modes of our hyperbolic lattice can be captured by the 1D hyperbolic band theory, by studying the **DOS comparisons** between 1D hyperbolic band theory and finite networks.

We are confident that the Reviewers will find the revised version of our manuscript and figures considerably improved, and that they will be able to express again a positive recommendation for publication. We provide below a point-by-point response to the comments made by all three Reviewers.

Reviewer 1

R1.1: The authors present a theoretical + experimental study of a non-reciprocal microwave scattering network with the geometry of the {8,3} hyperbolic lattice. Each node of the lattice supports a non-reciprocal "circulator" element which breaks time-reversal symmetry and allows for the possibility of topological phenomena such as Chern numbers and chiral edge states.

On the theoretical side, the authors first model an infinite scattering network using hyperbolic band theory. The system is defined by a unitary but not symmetric (on account of time-reversal breaking) S-matrix $S(\mathbf{k})$ where \mathbf{k} lives in a higher-dimensional Brillouin zone. The eigenvalues of this S-matrix play a role analogous to the quasi-energy band structure of a periodically driven (Floquet) system. Depending on the range of parameters for the circulators, this band structure supports band gaps whose topology can be characterized by momentum-space topological invariants, either Chern numbers or winding numbers discussed in the Floquet context. From the momentum-space analysis, the authors predict "anomalous Floquet insulator" phases with nontrivial winding numbers (but trivial Chern numbers) and Chern phases with nontrivial Chern numbers. The authors then investigate edge phenomena by imposing open (reflecting) boundary conditions in one direction in the band theory, or directly simulating a finite lattice with an edge. They find that band gaps with nontrivial winding number (not momentum-space Chern number) are associated with chirally dispersing modes in both the band theory and the real-space simulation.

Next, the authors test these predictions experimentally by both measuring edge propagation and also directly measuring the "bulk" topological invariant (winding number) directly. The latter is measured by introducing specific tunable phase elements that impose a phase difference across a "cut" through the device, and measuring the corresponding winding in the reflection coefficient, which detects spectral flow across the gap in the scattering resonances over a flux-pumping period.

In my opinion, this is a careful and detailed study that constitutes a significant advance in the field of hyperbolic lattices. From the experimental standpoint, it introduces microwave scattering networks as a new physical platform to study non-Euclidean phenomena in condensed matter physics, alongside circuit QED architectures and (low-frequency) electric circuit networks. It also investigates non-reciprocity in the hyperbolic context in the first time (although Ref. 17 did introduce time-reversal breaking complex phase elements), and demonstrates in a compelling manner the presence of chiral topology through both bulk and edge measurements. From the theoretical standpoint, it extends the study of topological phenomena in hyperbolic lattices to scattering networks.

I am happy to recommend the paper for publication but would first like the authors to address the following questions/comments.

Response: We thank the reviewer for his/her careful review, positive comment on our theoretical/experimental achievement, and the recommendation for publication.

R1.2: 1. The authors state that previous experimental work has left “unidirectional edge wave transport completely unobserved” in the hyperbolic context. However, Ref. 16 claims to have experimentally realized a hyperbolic version of the Haldane model and observed unidirectional propagation of a voltage package on the edge. The authors should better clarify the relationship of this work to theirs.

Response: We would like to emphasize the key difference that Ref. 16 realized two “spin” copies of hyperbolic Haldane models, whose experimental model is a reciprocal system with time reversal symmetry, allowing for pseudospin-dependent edge modes. Let us elaborate further on two main differences between our work and Ref. 16.

(1) The first difference lies in whether it is time-reversal invariant or not. The experimental sample of Ref.16 involves inductors and capacitors (i.e., LC resonator circuits as lattice sites and capacitors as couplings). Therefore, such an experimental platform preserves time reversal symmetry (if it is in the clean limit, there is no gain or loss), and is definitely reciprocal. To realize the edge modes, they first construct a pair of pseudospins $V_{\uparrow, i} = V_{i,1} + V_{i,2}e^{\pm i2\pi/3} + V_{i,3}e^{\mp i2\pi/3}$, with $V_{i,1}$, $V_{i,2}$ and $V_{i,3}$ defining the voltages on three nodes of a lattice sites. Then, by exciting one pseudospin, the pseudospin-dependent edge mode is observed. It is clear that the existence of these pseudospin-dependent edge modes relies on time reversal symmetry, and therefore cannot belong to class A. In practice, if there is any defect or disorder inducing coupling between pseudospins, such topological edge states would be quickly destroyed. In contrast, our hyperbolic scattering network breaks time reversal symmetry and reciprocity, presenting genuine unidirectional topological edge states, which are robust against defect or disorder.

(2) The second difference is the importance of wave behavior. The experimental demonstration of Ref.16 is conducted on lumped circuits, which operate in a quasi-static regime. These Kirchhoff’s laws governed circuits describe a discrete coupled-oscillator system with no spatial extent, and all the circuit starts reacts to external stimuli with no time delay. In contrast, our scattering network provides topological edge states associated with propagating GHz electromagnetic waves, exploring topological transport associated with hyperbolic wave physics. Furthermore, thanks to the phase delays imparted to waves as they travel, we can observe hyperbolic anomalous and Chern phases. We are able to study anomalous Floquet topology and demonstrate interesting properties such as the superior robustness of anomalous hyperbolic edge transport over the Chern one under phase delay disorder. This is not possible in a lumped circuit, which by definition does not involve propagation delays.

R1.3: 2. Likewise, it would be useful to have a brief discussion of time-reversal breaking vs non-reciprocity, given the experiment of Ref. 17 which had previously introduced complex phase elements in the hyperbolic context.

Response: Ref. 17 simulates an infinite hyperbolic lattice by adding a complex Bloch phase factor generated from a complex-phase circuit. We agree that this periodic boundary condition with Bloch phases definitely breaks time-reversal symmetry and reciprocity, as expected for any Bloch boundary condition, however it is applied to an infinite hyperbolic lattice that does not break time-reversal symmetry. In contrast, despite the fact that we use a similar boundary condition in our invariant measurement, our hyperbolic scattering network *itself* is time reversal asymmetric and non-reciprocal.

Revision: We have added a discussion on Ref. 17 to elucidate the distinction of time-reversal breaking and non-reciprocity between our work and Ref. 17.

R1.4. 3. In the last sentence of the section “Chiral edge modes in hyperbolic anomalous and Chern phases”, the authors write that “the grey line on the center ($\xi = \eta$) denotes a trivial insulator with all bandgaps closed.” If the bandgaps are closed how can this be an insulator? Do the authors mean a metallic or semi-metallic phase instead?

Response: Thank you for detecting this glitch. There is a point on this line that corresponds to a semi-metal, the rest is insulating.

Revision: We have revised the main text, which now reads “The grey line represents the trivial insulating phase, where its middle point ($\xi = -\eta = 0$) corresponds to a semi-metallic phase with all bandgaps closed, serving as a crossing point of three phases (i.e., anomalous, Chern, and trivial phases).”

R1.5: 4. At the end of the next section (Observation of hyperbolic chiral transport), the authors say that “the bulk modes scatter in all possible directions”. Yet from Fig. 3e it seems the scattering pattern is not completely isotropic but retains a bit of chirality. Do the authors have an explanation for this?

Response: This is due to the fact that nothing prevents the bulk modes to exhibit non-reciprocity to some extent, but in a diffusive manner, as depicted in Fig. R1. However, their non-reciprocity is not as pronounced or

controllable as that of topological edge modes, since they typically consist of a mixture of modes given the high density of states.

Revision: We have made a remark on this point in the main text when describing the profile of bulk modes.

Fig. R1 Measured field maps of bulk modes.

R1.6: 5. I eventually understood that the authors chose a longer CPW in the center of the device for purely geometric/engineering considerations (to have enough room to pack everything?) but I was initially confused by the two lengths. Although perhaps obvious to some, it might be useful to mention this point explicitly somewhere.

Response: Yes, this deliberate arrangement of the longer coplanar waveguides (CPWs) ensures that all CPWs, including the one on the extensive boundary, experience the same phase delay (at least within [5.75, 6.10] GHz).

Revision: We have further highlighted this important design consideration in the main text.

Reviewer 2

R2.1: The manuscript is an experimental study of the topological properties of models in non-Euclidean spaces. The Authors consider a planar projection of an hyperbolic lattice by coupling different scattering elements operating at microwave frequencies. In particular, the network connections between lattice sites in the hyperbolic space are realized via non-reciprocal circulators, where a phase delay is introduced. By varying two angular parameters of the circulator, setting different reflection and non-reciprocity degrees, a topological phase diagram is reconstructed. The findings are supported by the observation of unidirectional propagation on the edge of the system, associated with nonzero topological invariants, measured through a generalized Laughlin pumping.

The manuscript advances the field of topological physics, reaching out to new synthetic metamaterials on hyperbolic lattices, where experimental investigation are just begun. I believe that these results can be of immediate interest to the community working in the field, as its importance can stretch towards novel interplay of topology and possibly also non-Hermitian effects in non-Euclidean spaces.

Overall the paper is well written and organized, while the methodology and the extended information are sufficiently detailed and provide enough support to the arguments presented in main text. However, I have few comments that the Authors should address in order to increase the clarity of the paper and its conclusions.

Response: We thank the reviewer for his/her careful review and positive words about our work.

R2.2: 1) The band structure in Fig.2 is obtained with the phase of the circulator ϕ as a quasi-energy of a Floquet model. This analogy is formally evident in Eq.1 if the scattering matrix is the time-evolution operator of a temporally modulated Hamiltonian $H(t) = H(t+T)$ whose state is $|\psi(t+T)\rangle = e^{-i\epsilon T} |\psi(t)\rangle$, with ϵ being the quasi-energy. Is this analogy based on physical ground or is it just a formal one? What would be the temporal modulation of what quantity in your case?

Response: Indeed, this analogy is both formally and physically grounded. Our hyperbolic scattering network can be mapped into a cyclic oriented network describing a quantum walk, with $S(\mathbf{k})$ being the time-evolution operator of the system¹⁻⁵, as illustrated in Fig. R2. Due to the non-reciprocity of the three-port scattering nodes in the original network (Fig. R2a), the links (Fig. R2b) are now unidirectional, which force wave packets traveling in this oriented graph to experience scattering on the four-link vertices (S_1, S_2, S_3) in a sequential order. This cyclicity plays the role of time-ordering in the associated stepwise time-dependent system, and allows both the restauration of the notion of time-modulation and the exact mapping to a time-Floquet system. The order is determined by the connection condition in the oriented network. If the reviewer is interested in this analogy, we kindly suggest exploring the relevant references^{2,3} for a complete picture.

Revision: We have added this mapping to the Supplementary Note 2 and Fig. R2 as a new Supplementary Fig. 2.

Fig. R2 Mapping of the $\{8,3\}$ hyperbolic lattice to a cyclic oriented network.

R2.3: 2) Related to my previous comment, it is also not immediately evident to me why is the phase of the circulator ϕ used as energy in the band structure and not the actual energy/frequency. How would the plot Fig. 2d look like if the frequency was plotted instead of ϕ ? I expect some differences but at least the location of the gap-closing point signaling the topological phase transitions should exactly match.

Response: One is free to adopt both points of views: a phase-delay band structure assumes that the frequency is fixed, and the network links are scaled, whereas a frequency band structure assumes instead a fixed length of the network links and a varying frequency. However, it is clear that the underlying physics are better described by the Floquet point of view, since only the phase-delay band structure can allow the distinction between anomalous and Chern phases. Without this distinction, it is not possible to understand why some networks are robust to length disorder at a given frequency, and some others are not.

Nevertheless, we provide in Figs. R3a and R3c the frequency band structures for two different circulators. In this point of view, we can only compute band Chern numbers, and only define Chern or trivial phases. However, from the Floquet standpoint the edge states fall either in the anomalous (Fig. R3b) or Chern (Fig. R3d) phases, and will have very different resilience to disorder (please refer to Supplementary Note 6 and Supplementary Fig. 11 in the revised manuscript). The Floquet point of view is the only way to explain this physical difference by unveiling that the edge states actually belong to a different Floquet topological phase^{4,5}.

Fig. R3 Frequency band structures and phase-delay band structures of our hyperbolic network. a,c, Bulk frequency band structures for two different angular parameters $\zeta = -\eta = \pi/8$ (a) and $\zeta = -\eta = 7\pi/24$ (c) that characterize the circulators at frequency f_0 . b,d, Bulk phase-delay band structures for two different angular parameters at frequency f_0 , corresponding to anomalous and Chern phases.

R2.4: 3) Is the circulator breaking the time-reversal symmetry? I am thinking whether the Chern insulator phase belongs to class A or to some of the other 4-dimensional classes? Would be nice to have it written.

Response and revision: Yes, the circulators break the time-reversal symmetry, and both the anomalous and Chern phases belong to class A. Following the reviewer's suggestion, we now mention the class of Chern and anomalous phases in the revised version of manuscript.

R2.5: 4) At page 5, when discussing the location of the chiral edge modes, it is said that they are located "either at the top edge (red lines) or the bottom one (blue lines)". How are "top" and "bottom" edge defined here, since from Fig. 1 and 2c (but also Fig. 3e and so on) the real lattice is defined on a circle? I assume it has to do with the reflection boundary conditions along k_1 , but what is the meaning of propagation along k_1 in real space? It is hard to visualize otherwise.

Response: We agree that top and bottom are confusing notions when discussing the location of the chiral edge modes. For illustration, we show the profiles of chiral edge modes (i.e., red and blue lines in Figs. 2a-b) on a super cell in Fig. R4b. We can say that the edge modes are localized on two opposite sides of the boundary.

Due to the non-commuting translation generators of hyperbolic lattices, it is challenging to envision a hyperbolic network that is finite in one direction while remaining periodic in others, particularly in a Poincaré disk model. Hence, when referring to the super cell, we adopt k_i instead of specifying the direction in real space.

Revision: We thank the reviewer for this careful consideration. Accordingly, we have added the profiles of chiral edge modes as a new Supplementary Fig. 5, and revised the corresponding sentence to avoid confusion.

Fig. R4 Super cell with the profiles of chiral edge modes, corresponding to the red and blue markers in the band structure. The super cell contains three unit cells (split by the dashed lines), where the periodic boundary conditions (PBCs, green, orange

and blue lines) are employed along the k_2 , k_3 , and k_4 directions, and the full-reflection boundary conditions (FBCs, red lines) are applied along k_1 .

R2.6: 5) At page 3, the Authors say “Compared to the traditional case of Chern edge modes, anomalous edge modes are robust to any distributed disorder in the structure of the planar graph, with superior resistance to backscattering.” I don’t think this sentence is true, as chiral modes are either protected or not protected, so a topological invariant cannot protect them “more” than another. Authors should either rephrase the sentence, or provide evidence: for example, can this superior resistance to backscattering be seen from their experimental findings in Fig. 2 and 3?

Response: While A-class topological systems offer robust protection against local disorder and weak disorder throughout the entire system, this may only be true before the onset of Anderson localization. It is therefore important to note that for different systems, the disappearance of topological protection can occur at different levels of disorder, or even not occur at all^{4,5}. In the 2D Euclidean space, it was shown for instance that in the anomalous phase, trivial Anderson localization can surprisingly be completely avoided, whereas Chern phases behave as expected and trivially localize. The question we address here is: does this behavior remain true in hyperbolic cases?

To clear any doubt, let’s be quantitative and study the robustness of hyperbolic anomalous and Chern phases (Fig. R5). We consider a phase-disordered hyperbolic scattering network, where the disorder is introduced by adding randomly generated phase delays within $[-\delta\phi/2, \delta\phi/2]$ around $\phi = \pi/8$, varying $\delta\phi$. Each value on the solid line is averaged over 500 realizations of random disorder. Fig. R5a shows the average transmission between two ports on the boundary of the disordered network, with the dashed lines represents the first and last quartiles (Q1 and Q3). Remarkably, the chiral edge modes in the anomalous phase ($\xi = -\eta = \pi/8$) could survive under strong disorder strength up to the maximal value of 2π , in stark contrast to those in the Chern phase ($\xi = -\eta = 7\pi/24$), extending a result already known in Euclidean honeycomb lattices^{4,5}. One reason is that phase disorder in the anomalous phase introduces only fluctuations of quasi-energy that can locally switch the material between a bulk band and a topological gap, whereas a disordered Chern will be gradually “doped” with pieces of trivial insulators, eventually localizing every mode including the edge modes. Besides, we compute the average transmission along the edge in the phase-disordered network by varying $\xi = -\eta$ and disordered strength $\delta\phi$, as shown in Fig. R5b. This statistical analysis provides further evidence of the existence of an anomalous disordered phase, since a region of the parameters space remains blue regardless of disorder. Finally, we compute the average transmission in the presence of fully-random phase disorder (with a disorder strength of 2π) for any possible circulator (Fig. R5c). We confirm that a region of high average transmission persists in the anomalous phase, clearly evidencing the possibility for anomalous hyperbolic networks to survive arbitrarily large distributed phase disorder.

Fig. R5 Robustness of anomalous and Chern phases. a, Transmission between two ports on the boundary of a phase-disordered hyperbolic scattering network. The disorder is introduced by adding randomly generated phase delays within $[-\delta\phi/2, \delta\phi/2]$ around $\phi = \pi/8$, varying $\delta\phi$. Here, anomalous and Chern phases have the angular parameters of $\xi = -\eta = \pi/8$ and $\xi = -\eta = 7\pi/24$, respectively. Each value on the solid line is averaged over 500 realizations of random disorder. The dashed lines are the first and last quartiles (Q1 and Q3). b, Average transmission in the phase-disordered network as a function of $\xi = -\eta$ and disordered strength $\delta\phi$. c, Average transmission in the presence of fully-random phase disorder (with a disorder strength of 2π) at each point of the phase diagram.

Revision: We have added this discussion to the Supplementary Note 6 and Fig. R5 above as a new Supplementary Fig. 11.

R2.7: 6) The Authors use the Laughlin pumping argument to measure the winding number, and show that some bandgaps have zero winding for the Chern case, while the Chern number calculation is done via Wilson loop extraction. Would it be possible to measure some quantized bulk response to directly measure the Chern number?

Response: Bulk conductivity can be a local Chern marker, by using the local cross conductivity measured in the bulk of the system⁶⁻⁸. However, it is not quantized and it is not straightforward to measure. The only quantized and measurable topological invariant we know in the case of finite unitary systems is our topological scattering invariant, which uses probes to detect the topology of the system.

Reviewer 3

R3.1: In this manuscript, the authors designed and implemented a hyperbolic lattice in a planar non-reciprocal scattering network. They use hyperbolic band theory to compute band structure numerically and extracted the unitary homotopy invariant $W\varphi$ and Chern numbers from it. The numerical results show that their model could be in the anomalous phase or Chern phase depending on the parameter ξ and η and can support chiral edge waves in the anomalous phase. They further experimentally demonstrates the chiral edge waves and measured the unitary homotopy invariant $W\varphi$ based on the generalized Laughlin's argument they developed for the hyperbolic lattice. This is a novel experimental systems on hyperbolic lattice with breaking time reversal symmetry and the authors showed for the first time chiral edge states in hyperbolic lattice. Given the novelty above, I recommend the paper be published in Nature Communication after the following comments has been addressed.

Response: We thank the reviewer for his/her careful review, and the recommendation for publication.

R3.2: First, the authors computed the band structure of the model in 1D representations. However, there are two potential problems the authors need to explain due to the fact that the Fuchsian group has higher-dimensional irreducible representations and the hyperbolic lattice has a large boundary. First, the 1D irreducible representations only captures a small amount of bulk states, why can they give correct topological invariants that characterizes the boundary modes? Second, when cutting a finite piece from the hyperbolic lattice, the boundary to bulk ratio never tends to zero and the boundary effect can not be ignored. It is important to check that the band structure can actually reflect the properties of the system. In other words, it is worthwhile to compute the spectrum of the real space scattering matrix and compare them to the band structure to see at least if the DOS is similar between the 1D band theory and the real space finite lattice.

Response: We totally agree with the reviewer that in addition to the 1D irreducible representations, the higher dimensional irreducible representations could potentially contribute to the bulk modes. We also agree that the boundary effect of a finite hyperbolic network cannot be ignored, given that the ratio between the boundary and bulk never tends to zero in the thermodynamic limit.

To address the comment, we implement the reviewer's suggestion, and compare the total density of states (DOS) for a finite network to that obtained from the 1D hyperbolic band theory (HBT). We adopt the anomalous phase with angular parameters of $\xi = -\eta = \pi/8$ as an example. The DOS is defined as $\text{DOS}(\varphi) = N(\varphi)/\delta(\varphi)$, where $N(\varphi)$ counts the number of eigenmodes within $[\varphi, \varphi + \delta(\varphi)]$. The computed DOS for a super cell (Fig. R6a) is in agreement with that obtained from exact diagonalization for a closed finite network (Fig. R6b), both of which clearly identify the edge modes and bulk modes. This distinction is further confirmed by the skin distance for a closed finite network and the two-port edge transmission on an open finite network, as shown in Figs. R6c-d, respectively.

Fig. R6 Identification of chiral edge modes and bulk modes. (a) Density of states (DOS) obtained from the hyperbolic band theory (HBT) for a super cell. The super cell consists of 5 unit cells, with the periodic boundary conditions along k_2 , k_3 , and k_4 directions, and the full-reflection boundary conditions along k_1 . (b) DOS obtained from the exact diagonalization for a closed 4-layer network. (c) Skin distance for a closed 4-layer network. The skin distance is defined as $p_s(|b\rangle) = \sum_i p_s(i) |b_i|^2 / \sum_i |b_i|^2$, where $|b_i|$ is the amplitude of mode $|b\rangle$ at port i , and $p_s(i) \in \mathbb{N}$ is the skin index of port i , ranging from $p_s = 1$ on the boundary to a (size-dependent) maximal value at the center. (d) Two-port edge transmissions on an open 4-layer network. Here, we take the anomalous phase ($\xi = -\eta = \pi/8$) as an example.

Next, to test how well the 1D HBT describes the infinite lattice behavior, we compare the bulk DOS obtained from HBT for a unit cell, HBT for a super cell, and exact diagonalization for a finite network. In the case of the super cell and the finite network, the bulk DOS is determined by extracting the bulk modes using the skin distance, which effectively removes the boundary contribution to the total DOS. As shown in Fig. R7, the results show that a significant portion of the DOS remains in good agreement, giving us confidence that 1D HBT captures the bulk modes with sufficient accuracy in our hyperbolic scattering network.

Fig. R7 Bulk DOS comparison. (a) Bulk DOS obtain from HBT for a unit cell. (b-c) Bulk DOS obtain from HBT for a super cell (b) and exact diagonalization for a closed 4-layer network (c). The bulk DOS is determined by extracting the bulk modes using the skin distance with a threshold of 0.7.

Revision: We have added this discussion to the Supplementary Note 3 and Figs. R6-7 as a new Supplementary Figure 6-7.

R3.3: Second, the authors mentioned that the chiral edge waves are robust in Euclidean lattices. Are they robust in hyperbolic lattice as well? If they are, can the authors add some numerical or experimental results demonstrate this?

Response and revision: Yes, the hyperbolic chiral edge modes are robust against disorder, with the anomalous phase exhibiting superior resilience compared to the Chern phase. Please see the comment R2.6 for an extended quantitative study of the robustness of chiral edge modes in hyperbolic anomalous and Chern phases, which is now added in Supplementary Note 6 and Supplementary Fig. 11.

R3.4: Third, the authors numerically showed that their model can support Chern phase. However, they didn't show experimental phenomena associated to the non-trivial Chern vector. Can the author comment on why this part of the experiment is absent?

Response: Indeed, both Chern and anomalous edge states are observed experimentally, as shown in Fig. R8. We now show some experimental measurements in Supplementary Figs. 12a and 12d (revised version) that proves that the hyperbolic scattering network supports the Chern phase in the frequency band [5.0, 5.5] GHz.

Fig. R8 Measured field maps of chiral edge modes in both Chern and anomalous phases.

R3.5: Fourth, in page three paragraph 3, the authors mentioned that "These momentum components form a four-dimensional Brillouin zone, whose dimensionality is determined by the number of hyperbolic translation generators." This statement is inaccurate as the number of hyperbolic translation generators are not well defined for general hyperbolic lattice: the Fuchsian group, being a nonabelian group, can have different generating set with different number of generators. The precise statement should be "These momentum components form a four-dimensional Brillouin zone, whose dimensionality is determined by the minimal number of hyperbolic translation generators."

Response and revision: We thank the reviewer for this careful comment. Accordingly, we have modified the corresponding sentence in the main text.

R3.6: Fifth, the φ in Eq.(1) depends on k . It's better to explicitly mark this to avoid confusion.

Response and revision: Following the reviewer's suggestion, we have revised the equations in the text.

References:

1. Tauber, C. & Delplace, P. Topological edge states in two-gap unitary systems: a transfer matrix approach. *New J. Phys.* **17**, 115008 (2015).
2. Delplace, P., Fruchart, M. & Tauber, C. Phase rotation symmetry and the topology of oriented scattering networks. *Phys. Rev. B* **95**, 205413 (2017).
3. Delplace, P. Topological chiral modes in random scattering networks. *SciPost Phys.* **8**, 081 (2020).
4. Zhang, Z., Delplace, P. & Fleury, R. Anomalous topological waves in strongly amorphous scattering networks. *Sci. Adv.* **9**, eadg318 (2023).
5. Zhang, Z., Delplace, P. & Fleury, R. Superior robustness of anomalous non-reciprocal topological edge states. *Nature* **598**, 293–297 (2021).
6. d'Ornellas, P., Barnett, R. & Lee, D. K. K. Quantized bulk conductivity as a local Chern marker. *Phys. Rev. B* **106**, 155124 (2022).
7. Kitaev, A. Anyons in an exactly solved model and beyond. *Ann. Phys.* **321**, 2–111 (2006).
8. Mitchell, N. P., Nash, L. M., Hexner, D., Turner, A. M. & Irvine, W. T. M. Amorphous topological insulators constructed from random point sets. *Nat. Phys.* **14**, 380–385 (2018).

REVIEWERS' COMMENTS

Reviewer #1 (Remarks to the Author):

I have read the authors' response to my comments as well as the revised manuscript. The authors have addressed my comments satisfactorily. I support publication of the manuscript in its revised form.

Reviewer #2 (Remarks to the Author):

In the revised manuscript, the Authors have promptly responded to questions and comments, as well as criticisms, raised by all the Referees.

The Authors have made the suggested changes, taking the opportunity to improve the presentation and the clarity and significance of their findings. I recommend the publication of the manuscript in Nature Communication as is.

Reviewer #3 (Remarks to the Author):

In the response, the authors computed the total density of states (DOS) for a finite network and compare it to that obtained from the 1D hyperbolic band theory (HBT).

The result shows that the DOS obtained from HBT agrees with that computed for a finite network in the sense that they have gaps in the same range. The authors further shows that the states in the gap has lower skin distance and higher two port edge transmission. These results provide evidences that HBT is valid in describing the finite scattering network used in this experiment. Although an argument of why HBT works in this situation is still absent, their numerical results addresses my comment on the validity of the HBT.

The authors also provided evidence on the robustness of anomalous chiral edge states against randomly generated phase delays. Their detailed simulation addressed my comments on the robustness of the chiral anomalous edge states.

New experimental measurements in the new Supplementary Materials prove that the hyperbolic scattering network supports the Chern phase in the frequency band [5.0, 5.5] GHz, which addresses my third comment.

Overall, all my comments are properly addressed and I recommend the paper be published in Nature Communications.

Response letter, manuscript number NCOMMS-23-46090A

We thank all three Referees for their careful review and positive recommendation of our manuscript. We provide below a point-by-point response to the comments made by all three Reviewers.

Reviewer 1

R1.1: I have read the authors' response to my comments as well as the revised manuscript. The authors have addressed my comments satisfactorily. I support publication of the manuscript in its revised form.

Response: We thank the reviewer for his/her careful review and the recommendation for publication.

Reviewer 2

R2.1: In the revised manuscript, the Authors have promptly responded to questions and comments, as well as criticisms, raised by all the Referees.

The Authors have made the suggested changes, taking the opportunity to improve the presentation and the clarity and significance of their findings. I recommend the publication of the manuscript in Nature Communication as is.

Response: We thank the reviewer for his/her careful review and the recommendation for publication.

Reviewer 3

R3.1: In the response, the authors computed the total density of states (DOS) for a finite network and compare it to that obtained from the 1D hyperbolic band theory (HBT).

The result shows that the DOS obtained from HBT agrees with that computed for a finite network in the sense that they have gaps in the same range. The authors further shows that the states in the gap has lower skin distance and higher two port edge transmission. These results provide evidences that HBT is valid in describing the finite scattering network used in this experiment. Although an argument of why HBT works in this situation is still absent, their numerical results addresses my comment on the validity of the HBT.

The authors also provided evidence on the robustness of anomalous chiral edge states against randomly generated phase delays. Their detailed simulation addressed my comments on the robustness of the chiral anomalous edge states.

New experimental measurements in the new Supplementary Materials prove that the hyperbolic scattering network supports the Chern phase in the frequency band [5.0, 5.5] GHz, which addresses my third comment.

Overall, all my comments are properly addressed and I recommend the paper be published in Nature Communications.

Response: We thank the reviewer for his/her careful review and the recommendation for publication.